# Optimization, Kinetics, Thermodynamic and Arrhenius Model of the Removal of Ciprofloxacin by Internal Electrolysis with Fe-Cu and Fe-C Materials

**Tra Huong Do** [1,*], **Xuan Linh Ha** [2], **Thi Tu Anh Duong** [1], **Phuong Chi Nguyen** [1], **Ngoc Bich Hoang** [3] and **Thi Kim Ngan Tran** [3,4,*]

1   Chemistry Faculty, Thai Nguyen University of Education, Thai Nguyen 250000, Vietnam; anhttd@tnue.edu.vn (T.T.A.D.); phuongchi0715@gmail.com (P.C.N.)
2   International School, Thai Nguyen University, Thai Nguyen 250000, Vietnam; haxuanlinh@tnu.edu.vn
3   Institute of Environmental Technology and Sustainable Development, Nguyen Tat Thanh University, Ho Chi Minh 700000, Vietnam; bichhn@ntt.edu.vn
4   Faculty of Food and Environmental Engineering, Nguyen Tat Thanh University, Ho Chi Minh 700000, Vietnam
*   Correspondence: huongdt.chem@tnue.edu.vn (T.H.D.); nganttk@ntt.edu.vn (T.K.N.T.); Tel.: +84-392-073-898 (T.H.D.); +84-765-712-086 (T.K.N.T.)

**Abstract:** The ciprofloxacin (CIP) removal ability of a Fe-Cu electrolytic material was examined with respect to pH (2–9), time (15–150 min), shaking speed (100–250 rpm), material mass (0.2–3 g/L), temperature (298, 308, 323) and initial CIP concentration (30–200 mg/L). The Fe-Cu electrolytic materials were fabricated by the chemical plating method, and Fe-C materials were mechanically mixed from iron powder and graphite. The results show that at a pH value of 3, shaking time 120 min, shaking speed 250 rpm, a mass of Fe-Cu, Fe-C material of 2 g/L and initial CIP concentration of 203.79 mg/L, the CIP removal efficiency of Fe-Cu material reached 90.25% and that of Fe-C material was 85.12%. The removal of CIP on Fe-Cu and Fe-C materials follows pseudo-first-order kinetics. The activation energy of CIP removal of Fe-Cu material is 14.93 KJ/mol and of Fe-C material is 16.87 KJ/mol. The positive ΔH proves that CIP removal is endothermic. A negative entropy of 0.239 kJ/mol and 0.235 kJ/mol (which is near zero and is also relatively positive) indicated the rapid removal of the CIP molecules into the removed products.

**Keywords:** internal microelectrolysis; Fe-Cu and Fe-C; remove; ciprofloxacin; aqueous solution

## 1. Introduction

Antibiotic resistance, a serious problem to global health, is currently increasing to dangerous levels, especially in developing countries [1,2]. Since antibiotic residue is one of the reasons that cause antibiotic resistance, significant efforts have been devoted to developing techniques for the removal of antibiotic residue from water environments. Techniques that hold potential in the removal of antibiotics are diverse and might include adsorption [3], photocatalytic removal or advanced oxidation [4], water-soluble polymers [5–7]. Internal micro electrolysis is a technique that has been gaining popularity recently in the pretreatment process of wastewater, especially industrial wastewater. This method has been frequently applied to treat wastewater containing persistent organic substances with high concentrations of pollutants, usually in wastewater containing textile dyes [8–10], pharmaceuticals [11,12], paper chemicals [13], coal gasification [14], Cu-EDTA complexes [15], nitrogen compounds from sewage [16], electroplating and mechanical industrial discharges [17–21], oily substances [22] and TNT and RDX explosives [23].

The process of internal electrolysis commences when two materials, having different electrode potentials, contact and form a pair of microelectrodes. For Fe-C, Fe-Cu systems, iron plays the role of anode and copper or carbon is the cathode, which is similar to the

pair of micro-batteries in metal corrosion. With a pair of micro-batteries with a potential of about 1.2 V, a small current of about $\mu A$ appears, acting as a redox agent in the removal reaction of organic compounds adsorbed on the electrode surface. Due to such a principle, the process of Fe-C and Fe-Cu microelectrolysis is also called internal microelectrolysis. Therefore, it is possible to dissolve iron without using an external current by establishing microbatteries in the form of Fe-C or Fe-Cu composite materials, which is an important advantage in the internal engineering of wastewater pretreatment electrolysis [8–27]. The reactions that take place during electrolysis are as follows:

$$\text{Reaction at the anode (oxidation): Fe} \rightarrow Fe^{2+} + 2e \tag{1}$$

$$\text{Reaction at the cathode (reduction): } 2H^+ + 2e \rightarrow 2[H] \rightarrow H_2 \tag{2}$$

When the oxygen is presented: acid conditions [24]:

$$O_2 + 4H^+ + 4e \rightarrow 2O + 4[H] \rightarrow 2H_2O \tag{3}$$

In addition, the removal of organic compounds in the internal electrolysis method is also thought to be based on the reduction of iron at valence, the reduction of [H], the oxidation of $\cdot O$, the formation of complexes, the colloidal accumulation of iron ions and the adsorption of iron hydroxide [25].

According to Yang [25], Fe-C or Fe-Cu materials embedded in the electrolyte will form corrosive batteries with electronegative metal corroded from the anodic reaction: $Me \rightarrow Me^{n+} + ne$. Corresponding to the anodic process is the cathode depolarization process on the more electropositive part, such as releasing $H_2$ gas in an acidic environment, depolarizing dissolved oxygen or other substances or ions. In the presence of $Fe^{2+}$ and $H_2O_2$ in the electrolyte, there can also be a Fenton reaction to form $\cdot OH$ radicals. If in the solution the organic substances RX (organochlorine compound) and $RNO_2$ (aromatic nitrocyclic compound) are components capable of accepting electrons on the anode surface (Fe) to transfer to the cathode, they are deduced according to dechlorination and deaminoation reactions. Then the pollutant will be converted into non-toxic or less toxic intermediate products, which are easier to biodegrade. Therefore, in terms of chemical and physicochemical natures, the process of internal electrolysis will be different from the process of non-valent iron in the treatment of pollutant waste (producing A-sized internal electrolyte and producing radicals, OH).

Based on the above-mentioned advantages and analysis, the method of internal electrolysis is regarded as an efficient and affordable technology for decomposing antibiotics in aqueous media. There have been many studies on Fe-Cu and Fe-C materials; however, research on CIP removal on these two materials is still limited, and no author has fully studied the dynamic process, thermodynamics or activation energy of CIP removal of Fe-Cu and Fe-C. In order to improve the efficiency of antibiotic treatment in this paper, we studied the influence of factors, such as pH, treatment time, material weight, shaking speed, concentration and temperature, on the yield, study of kinetics, thermodynamics and Arrhenius model in the removal of ciprofloxacin in aqueous solution of Fe-C and Fe-Cu electrolytic materials.

## 2. Materials and Methods

### 2.1. Producing Fe-C and Fe-Cu Materials

2.1.1. Producing Fe-C Materials

A mixture was formed with the following ingredients: 95% Fe, 3% graphite and 2% bentonite binder additive. The material was dried at 80–105 °C for 2 h, then underwent mass sintering at 500–600 °C for 4 h, followed by natural cooling. The material was then stored in a desiccator for further studies [26].

### 2.1.2. Producing Fe-Cu materials

Fe powder with a size of less than 50 μm and purity of 99.9% (PA, China) was soaked in 30% NaOH solution for 10 min to remove grease and clean the surface. The surface was activated by treatment in 7.4% HCl for 3 min. A diluted HCl solution was prepared with 37% HCl solution. The material was then washed several times with water, dried at 105 °C for 2 h, then allowed to cool and stored in a sealed glass jar. The Fe-Cu sample was fabricated by chemical plating in 5% $CuSO_4$ solution (wt%). To be specific, a total of 100 g of Fe powder was added to 1 L of 5% $CuSO_4$ solution for a period of 2 min. The mixture was then washed several times with water and dried at 105 °C for 3 h under $N_2$. The material was then stored in a desiccator for further study [27].

### 2.1.3. Investigation of Surface Characteristics, Structure and Chemical Composition of Fe-C and Fe-Cu Electrolytic Materials

The surface morphological characteristics of Fe-C materials and Fe-Cu were determined using scanning electron microscopy (SEM). The material structure was analyzed by X-ray diffraction (XRD). The material composition was analyzed by energy dispersive spectroscopy (EDX). The specific surface area of the materials was analyzed by $N_2$ adsorption-desorption isotherm (BET). The results are shown in documents [26,27].

### 2.2. Ciprofloxacin

Ciprofloxacin (CIP) is an antibiotic of the 4 quinolone group of the 2nd generation of the Fluoroquinolone antibiotic system [28].

Molecular formula: $C_{17}H_{18}FN_3O_3$.

Molar Mass: 331.346.

The UV-Vis spectrum of CIP in aqueous solution is shown in Figure 1.

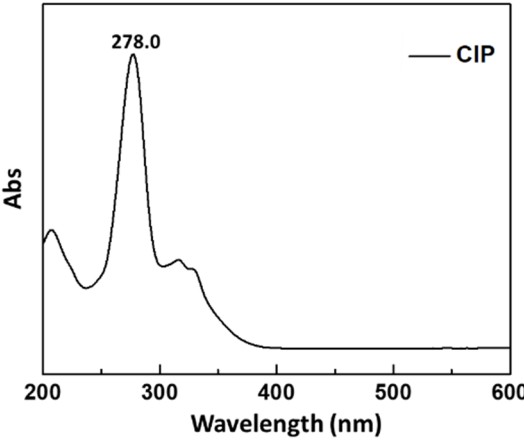

**Figure 1.** UV-Vis Spectrum of CIP, concentration 10 mg/L, pH = 3.

### 2.3. CIP Removal

The experimental factors, such as solution pH, time, the weight of Fe-Cu and Fe-C materials, shaking speed, initial CIP concentration and heat, were surveyed to evaluate the process of CIP removal. To ensure repeatability, each experiment was performed at least 3 times under the same conditions. The results are presented as the average of the 3 experiments. Briefly, a certain amount of Fe-Cu and Fe-C materials was introduced into each 100 mL Erlenmeyer flask, followed by the addition of a certain amount of CIP solution. A total of 1 mol/L NaOH and 0.1 mol/L HCl were used to adjust the pH of the CIP solution. The flasks were shaken on a shaker with conditions varying depending on the experiment Table 1).

**Table 1.** Parameters in the experiments.

| Experiment/Parameters | pH | Time (min) | Material Mass (g) | Shaking Speed (rpm) | CIP Initial Concentration (mg/L) | Temperature (K) |
|---|---|---|---|---|---|---|
| Effect of pH | 2 to 9 | 120 | 0.1 | 250 | 50 | 298 |
| Effect of time | 3 | 15–180 | 0.1 | 250 | 50 | 298 |
| Effect of mass of initial materials | 3 | 120 | 0.01–0.15 | 250 | 50 | 298 |
| Effect of shaking speed | 3 | 120 | 0.1 | 100–250 | 50 | 298 |
| Effect of CIP initial concentration | 3 | 120 | 0.1 | 250 | 50 | 298 |
| Effect of temperature | 3 | 120 | 0.1 | 250 | 100 | 298, 308, 323 |

The concentrations of CIP before and after treatment with Fe-Cu and Fe-C materials were determined by ultraviolet-visible spectroscopy measured on Hitachi UH5300 machine at University of Medicine and Pharmacy—University of Science and Technology Thai Nguyen. They were measured with a wavelength range from 190 to 1100 nm, a scanning speed from 10 to 6000 nm/s, a wavelength accuracy of $\pm 0.3$ nm and noise <0.0001 nm.

The CIP removal efficiency was calculated according to the following formula:

$$H\% = \frac{(C_0 - C_t)}{C_0} \times 100\% \tag{4}$$

where $C_0$ is the initial concentration of CIP solution before removal (mg/L), $C_t$ is the concentration of CIP solution after removal (mg/L) and H is the removal efficiency (%).

The Arrhenius formula is:

$$\ln k = -\frac{E_a}{RT} + \ln A_o \tag{5}$$

where k is the reaction rate constant.

$E_a$ (KJ/mol) is the Arrhenius activation energy.

$A_o$ is the factor that has the same value and has the same units as k.

The enthalpy change of the CIP removal process could be identified as the $E_a$-value, which could be calculated by the following equation with the assumption that 1 mol of CIP molecules was degraded [29].

$$\Delta H^0 = E_a + RT \tag{6}$$

After obtaining $E_a$, the entropy change of the reaction could be calculated from the following equation, in which the $K_B$ (Boltzmann constant) and h (Plank constant) are known.

$$\ln(k/T) = \ln(K_B/h) + \Delta S^0/RT - (\Delta H^0/RT) \tag{7}$$

Plotting $\ln(k/T)$ against $(\Delta H^0/RT)$ yielded an intercept value that is interpreted as $\Delta S^0$ and is used for the calculation of the activation Gibbs free energy as follows [29]:

$$\Delta G^0 = \Delta H^0 - T\Delta S \tag{8}$$

## 3. Results and Discussion

### 3.1. Effect of Initial pH on CIP Removal

From Equations (1) and (2), it is shown that the pH value has a great influence on the reaction rate and the redox's ability to generate [H]. When the pH becomes more acidic, the amount of $H^+$ provided for the reaction is enough or in excess and can make the rate of electrolysis faster. As the initial pH value becomes lower, the [H] concentration becomes higher. Moreover, in the presence of $O_2$, the cathode reduction of the internal electrolysis reaction can also occur according to the following reaction:

$$O_2 + 4H^+ + 4e \rightarrow 2O\cdot + 4[H] \rightarrow 2H_2O; \quad E_0 (O_2/H_2O) = 1.23 \text{ V} \tag{9}$$

Thus, more $H^+$ will produce more [H] and O, leading to the higher redox capacity of CIP and achieving better CIP treatment efficiency.

On the other hand, the reactions of Fe in solutions with different pH can be represented by the following equations [25]:

$$Fe \rightarrow Fe^{2+} + 2e \tag{10}$$

$$Fe + 2H^+ \rightarrow Fe^{2+} + H_2 \tag{11}$$

$$Fe^{2+} \rightarrow Fe^{3+} + e \tag{12}$$

According to the Nernst Equation, the reducing ability of $Fe^{2+}/Fe$ will increase as the pH decreases. The initial pH value also affects the rate of corrosion reactions of Fe-C; Fe-Cu materials to form $Fe^{2+}$, $Fe^{3+}$, $Fe(OH)_2$ and $Fe(OH)_3$. In a more acidic environment than $Fe^{2+}$, $Fe^{3+}$ is easily formed, but it is difficult to precipitate $Fe(OH)_2$ and $Fe(OH)_3$. On the contrary, when the pH is high, the acidity decreases and the presence of dissolved oxygen will easily form $Fe(OH)_2$ and $Fe(OH)_3$, and the concentration will be gradually increased with the reaction time. Iron hydroxides are also indirect agents for the partial removal of CIP, as well as intermediate compounds of the treatment process by adsorption, coagulation and precipitation.

As shown in Figure 2, CIP removal under treatment with Fe-C and Fe-Cu materials decreases with increasing pH value. With Fe-Cu materials, the results show that the CIP removal efficiency increases when the pH increases from 2 to 3, eventually peaking at pH = 3 with a removal efficiency of 93.97%. In the increasing pH range from 3 to 9, the efficiency decreases again; at pH = 9, the removal efficiency was only 56.29%. Similar to Fe-Cu materials, Fe-C materials peak when investigating pHs in the range from 2 to 9, the highest removal efficiency was 88.90%, achieved at pH = 3, when the pH gradually approached the basic environment, the removal efficiency decreased gradually. This is explained by the reduced concentration of $Fe^{2+}$ and $Fe^{3+}$ ions as the pH increases, leading to a decrease in CIP removal by oxidation-reduction, chemical reactions and flocculation. However, due to the adsorption of activated carbon in Fe-C material [25], the CIP removal efficiency of this material is reduced compared to that of Fe-Cu at pH values $\leq 3$. As a result, we chose pH = 3 as the optimal pH value for CIP treatment of both Fe-Cu and Fe-C materials.

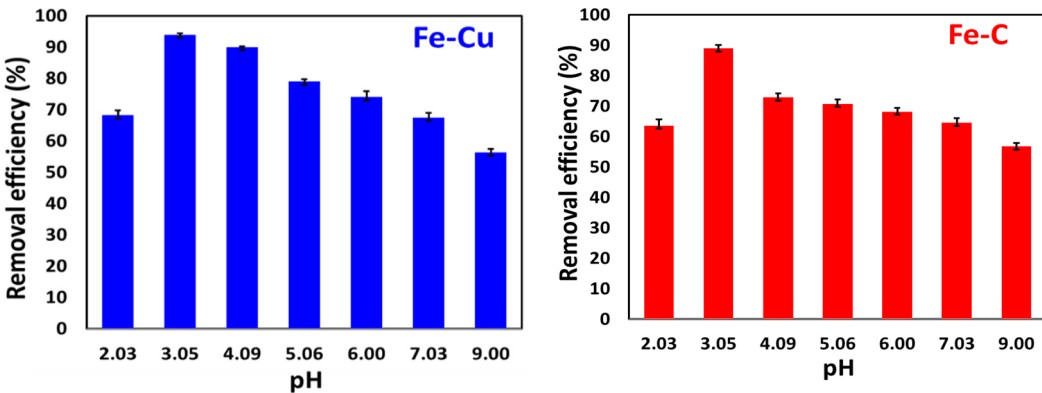

**Figure 2.** Effect of initial pH on CIP removal under Fe-Cu and Fe-C treatment.

### 3.2. Effect of Reaction Time on CIP Removal

Figure 3 shows that at pH = 3, when investigating at a time interval of 15–120 min, the CIP removal efficiency of Fe-Cu and Fe-C both increased rapidly. For Fe-Cu material, the CIP removal efficiency increased from 64.52% to 94.39%, higher than that of Fe-C material, which increased from 52.75% to 91.68%. During the next period of 120–180 min, the removal efficiency of both materials increased quite slowly (about 1%/30 min). This can be explained as follows: as the reaction time increases, the concentration of [H], $Fe^{2+}$ and $Fe^{3+}$ generated in the electrochemical reaction increases and the CIF content decomposed more. When the reaction time is too long, the CIP removal rate of the internal electrolytic

material will be slower, possibly because a large amount of $H^+$ is consumed along with the increase in the reaction time, leading to a decrease in [H] and increased oxygen content in the aqueous material, which increased the amount of $OH^-$ participating in the reaction with $H^+$, thereby reducing [H] [27]. Therefore, the time of 120 min was chosen as the optimal time for CIP processing. Based on the results in Figure 4, it can be seen that at the same initial investigation conditions, Fe-Cu materials have a better CIP removal efficiency than Fe-C materials.

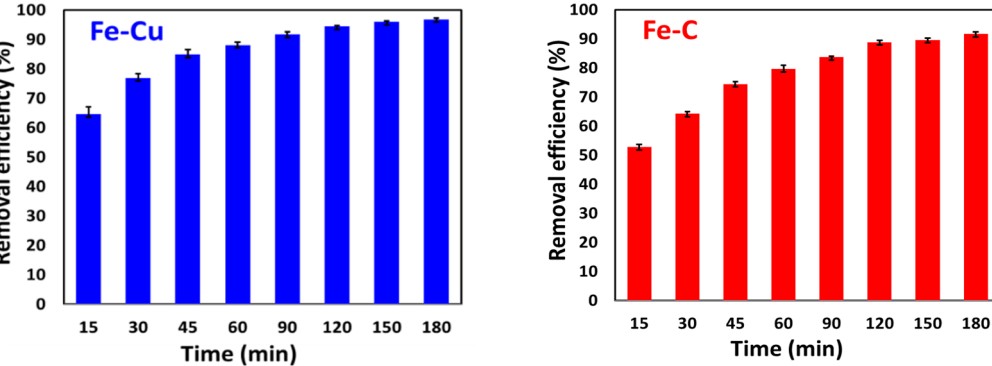

**Figure 3.** Effect of reaction time on CIP removal under Fe-Cu and Fe-C treatment.

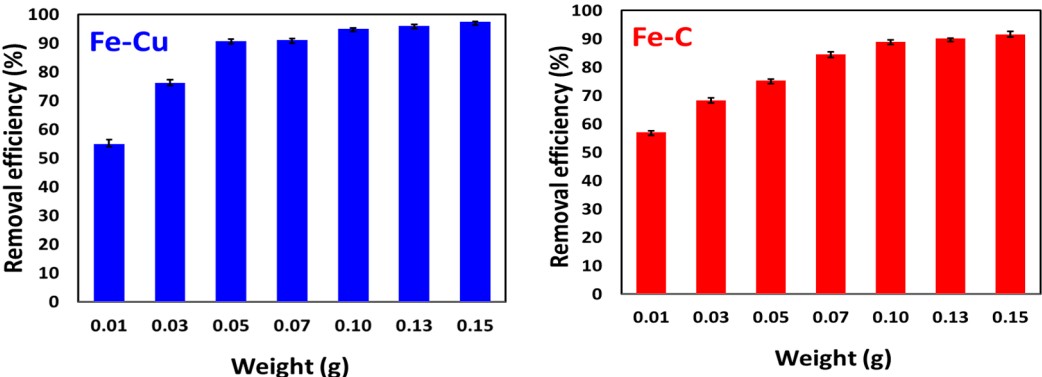

**Figure 4.** Effect of material mass on CIP removal under Fe-Cu and Fe-C treatment.

*3.3. Effect of Material Mass on CIP Removal*

When the mass of Fe-Cu increased from 0.01 to 0.10 g, the CIP removal efficiency increased sharply (54.97–95.01%). When the material weight increased from 0.10 to 0.15 g, the CIP removal efficiency increased more slowly and insignificantly (about 1%), which is similar to Fe-C materials. However, under the same initial material mass conditions, the CIP removal treated under Fe-Cu materials is always higher than that of Fe-C materials. This can be explained as follows: when the amount of Fe-C and Fe-Cu increased, more microbatteries were formed, which will lead to an increase in current efficiency during internal electrolysis. Therefore, the CIP removal efficiency increased. However, the sufficiently large quantity of iron existing in the material can inhibit electrode reactions. Therefore, the removal of contaminants should resort to the action of noncovalent iron [30], which holds a weaker removal capacity than the internal electrolysis [31]. Therefore, we choose the optimal mass of Fe-C and Fe-Cu materials for the CIP removal to be 0.1 g or 2 g/L to conduct further studies.

*3.4. Effect of Shaking Speed*

The shaking speed increases the dissolved oxygen content of the solution and the possibility of pollutant diffusion to the Fe-C and Fe-Cu electrode contact surfaces, as well as the rapid dispersion of the treated products at the electrodes polar into the solution. However, under a low pH, the dissolved oxygen content is less than in an alkaline

medium. The influence of shaking speed on CIP removal efficiency can be explained by the following reasons:

(1) The contact ability between the liquid phase (CIP solution)–solid phase (Cu and C cathode): the higher the shaking speed is, the larger the amount of CIP solution in contact with the cathode and vice versa. In addition, the faster shaking speed also increases the transport speed of the CIP solution and intermediate removal products to participate in the reaction between the (liquid) phase and the solid phase.

(2) Increasing the dissolved oxygen content in the solution: the higher the shaking speed is, the greater the dissolved oxygen content in the reaction solution is. In other words, oxygen in the air readily dissolves into the reaction solution under conditions of high shaking speeds. The process of internal electrolysis is the process of self-corrosion of iron at the anode (sacrificial electrode); then if there is oxygen at the cathode, the oxygen will be reduced and combined with the $H^+$ available in the acidic environment to produce $H_2O_2$. This agent then meets the nascent $Fe^{2+}$, where the Fenton reaction occurs in a pH of 2 to 3 to generate ·OH radicals, those ·OH radicals are capable of strongly oxidizing CIP. Therefore, CIP is not only reduced on the surface of the Cu and C cathodes but also enhanced by the Fenton mechanism in an acidic environment during internal electrolysis.

Figure 5 shows that when the shaking speed is higher, the CIP removal efficiency of Fe-Cu and Fe-C materials also increases. Therefore, we chose a shaking speed of 250 rpm as the optimal speed for the next studies.

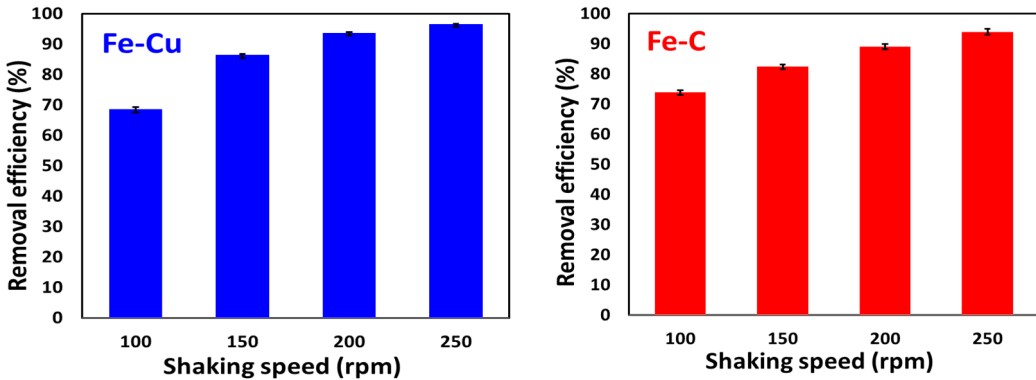

**Figure 5.** Effect of shaking speed on CIP removal under Fe-Cu and Fe-C treatment.

### 3.5. Effect of Initial Concentration CIP

As shown in Figure 6, when the initial CIP concentration is 33.44 mg/L, the CIP removal efficiency of both Fe-Cu and Fe- C materials is greater than 95.5%. When increasing the CIP concentration from 51.19 to 203.79 mg/L, the CIP removal efficiency of Fe-Cu and Fe-C materials both decreased gradually. With an initial CIP concentration of 203.79 mg/L, the CIP removal efficiency of Fe-Cu materials was 90.25% and Fe-C materials was 85.12%. This can be explained as follows: CIP removal depends on oxidation reactions by free radicals, mainly ·HO (Figure 6). At low CIP concentrations, the reaction rate will be less constrained by concentration. Furthermore, the lifetime of ·HO is too short to participate in the reaction with CIP. As the initial CIP concentration is gradually increased, the probability of a reaction between CIP and HO* increases, so the removal efficiency of CIP increases. However, at a high concentration of CIP solution, the mass of the material remains the same, the amount of ·HO formed in the internal electrolysis method decreases, so the CIP removal efficiency decreases [9]. However, the results from Figure 6 also show that Fe-Cu and Fe-C materials still give high CIP removal efficiency when the concentration is large.

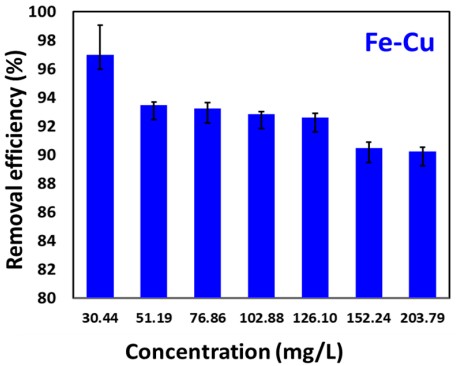 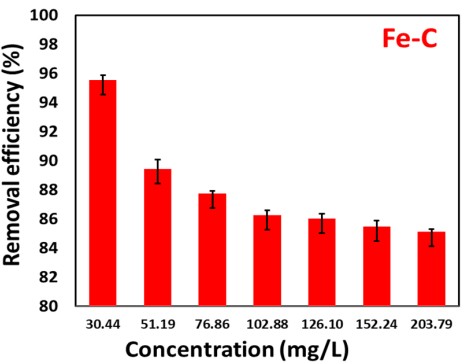

**Figure 6.** Effect of initial concentration CIP on CIP removal under Fe-Cu and Fe-C treatment.

*3.6. Effect of Reaction Temperature on CIP Removal*

The survey results show that, for both Fe-Cu and Fe-C materials, when the temperature increases, the CIP removal efficiency increases. This can be explained as follows: as the temperature increases, the reaction rate increases, the $\cdot$HO radical concentration increases, so the CIP removal efficiency increases. Figure 7 shows that the CIP removal reaction is endothermic.

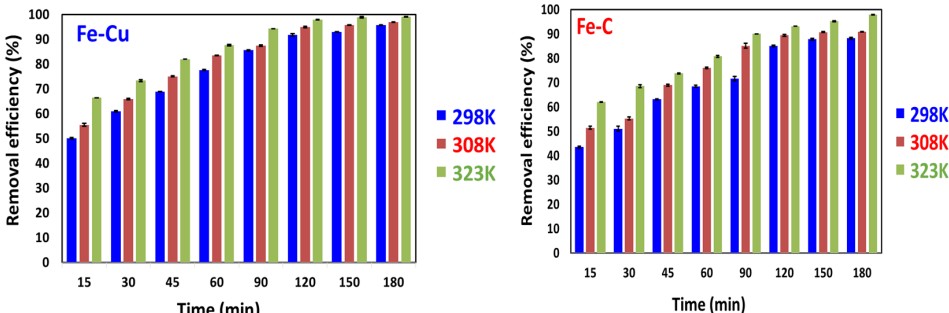

**Figure 7.** Effect of reaction temperature on CIP removal under Fe-Cu and Fe-C treatment.

*3.7. Kinetics of CIP Removal Treated by Fe-C and Fe-Cu Materials*

For the results of the influence of time on the CIP removal efficiency, we conducted a kinetic investigation of the CIP removal process according to the apparent kinetic equations of first, second and third as follows:

$$\text{The first apparent kinematics equation: } \ln C_t = - k_1.t + A_1 \tag{13}$$

$$\text{The second apparent kinematics equation: } 1/C_t = k_2.t + A_2 \tag{14}$$

$$\text{The third apparent kinematics equation: } 1/C_t^2 = 2k_3.t + A_3 \tag{15}$$

where $k_1$, $k_2$ and $k_3$ are the first, second and third-order reaction rate constants, respectively, and $A_1$, $A_2$, and $A_3$ are constants.

The results of the first, second and third order kinetic equations of CIP removal in aqueous solution of Fe-Cu and Fe-C materials are shown in Figure 8.

The results of Table 2 show that the CIP removal process at 298, 308 and 323 K of both Fe-Cu and Fe-C materials follows the apparent first-order kinetic model because the correlation coefficient of $R^2$ is high (>0.95), compared to the second (>0.8094)) and third-order (>0.5879) kinetic models.

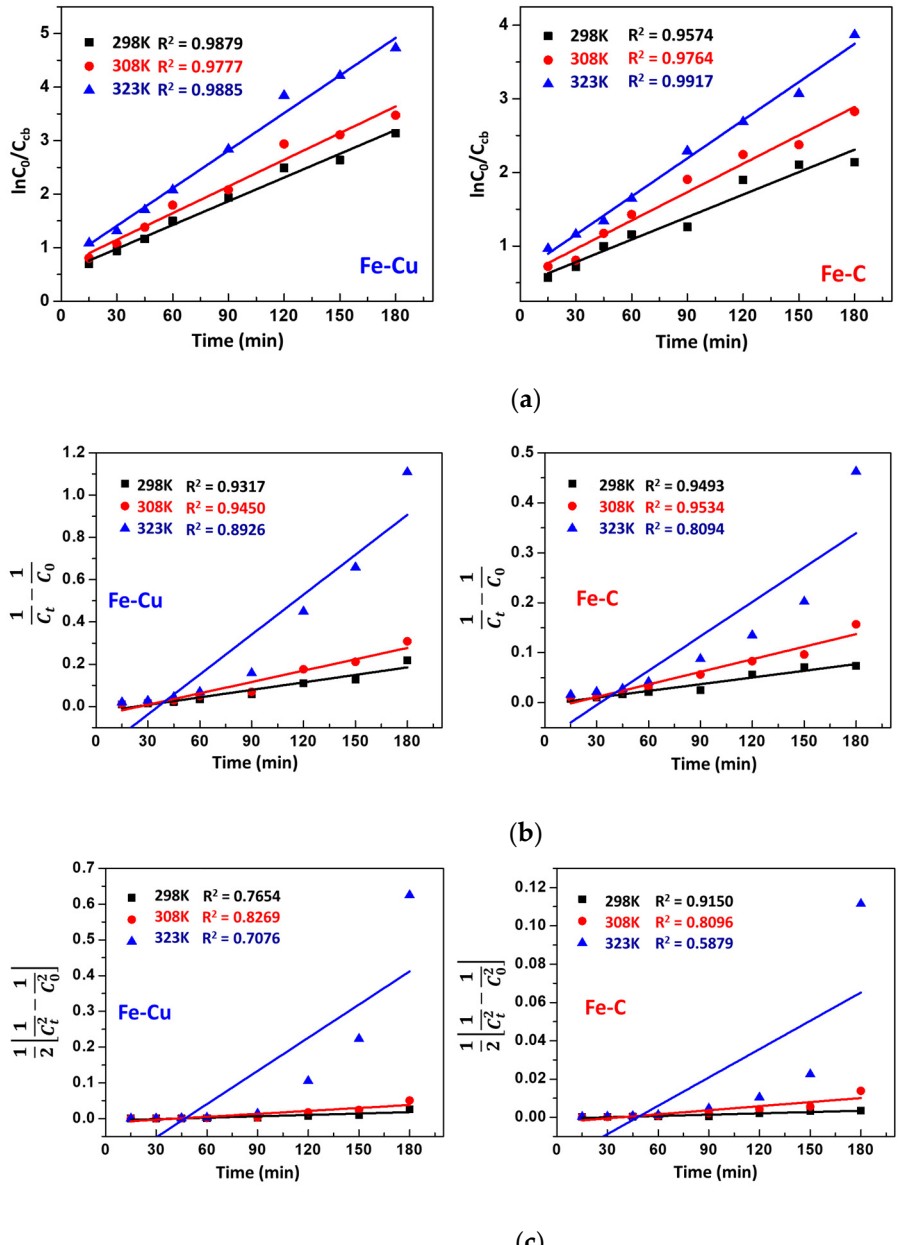

**Figure 8.** (**a**) The apparent first-order kinetic equation of Fe-Cu and Fe-C materials at 298, 308 and 323 K; (**b**) The apparent second-order kinetic equation of Fe-Cu and Fe-C materials at 298, 308 and 323 K; (**c**) The apparent third-order kinetic equation of Fe-Cu and Fe-C materials at 298, 308 and 323 K.

The CIP removal reaction rate constants of Fe-Cu and Fe-C materials both decrease with increasing temperature. This once again confirms that the CIP removal reaction of the two materials is endothermic.

At the same temperature, the CIP removal reaction rate constant of Fe-Cu material is larger than that of Fe-C material. This is consistent with the previous survey results; Fe-Cu materials showed higher CIP removal efficiency than Fe-C materials.

The results of calculating the amount of $E_a$ activation are based on the relationship between lnk and 1/T (Figure 9a)

**Table 2.** The calculation results of first-order, second-order and third-order CIP removal reaction rate constants and linear regression coefficients of Fe-Cu and Fe-C materials at different temperatures.

| First-Order Kinetic Equation Removal of CIP | | | | |
|---|---|---|---|---|
| | **Fe-Cu** | | **Fe-C** | |
| **T (K)** | **k (min$^{-1}$)** | **R$^2$** | **k (min$^{-1}$)** | **R$^2$** |
| 298 | 0.0148 | 0.9879 | 0.0102 | 0.9574 |
| 308 | 0.0166 | 0.9777 | 0.0129 | 0.9764 |
| 323 | 0.0234 | 0.9885 | 0.0173 | 0.9917 |

| Second-order kinetic equation removal of CIP | | | | |
|---|---|---|---|---|
| | **Fe-Cu** | | **Fe-C** | |
| **T (K)** | **k (min$^{-1}$·L·mg$^{-1}$)** | **R$^2$** | **k (min$^{-1}$·L·mg$^{-1}$)** | **R$^2$** |
| 298 | 0.0012 | 0.9317 | 0.0004 | 0.9493 |
| 308 | 0.0018 | 0.9450 | 0.0008 | 0.9534 |
| 323 | 0.0063 | 0.8926 | 0.0023 | 0.8094 |

| Third-order kinetic equation removal of CIP | | | | |
|---|---|---|---|---|
| | **Fe-Cu** | | **Fe-C** | |
| **T (K)** | **k (min$^{-1}$·(L·mg)$^{-2}$)** | **R$^2$** | **k (min$^{-1}$·(L·mg)$^{-2}$)** | **R$^2$** |
| 298 | 0.0001 | 0.7654 | 0.00002 | 0.9150 |
| 308 | 0.0003 | 0.8269 | 0.00007 | 0.8096 |
| 323 | 0.0031 | 0.7076 | 0.0005 | 0.5879 |

**Figure 9.** (**a**) Relationship between lnk and 1/T; (**b**) Relationship between ln(k/T) and 1/T.

As shown in Table 3, the activation energy of CIP removal of Fe-C materials is higher than that of Fe-Cu materials, showing that Fe-Cu materials decompose CIP better than Fe-C materials. This can be explained as follows: microscopic bimetallic electrolytic materials prepared by deposition, plating the second transition metals on the iron surface can greatly enhance the reduction of pollutants [32,33]. Previous studies showed that transition metals, such as Ni, Pd, Cu, Co, Au and Ru, could enhance the catalytic reactivity of Fe$^0$ [34]. The catalytic mechanisms of bimetallic internal electrolytic materials could be elaborated via two main explanations. First, bimetallic internal electrolysis and transition metal additives might promote the generation of atomic hydrogen ([H]abs), which might be absorbed on the surface of the material and induce indirect reduction [35,36]. Second, surface additives (i.e., metal transitions) might accelerate Fe$^0$ oxidation by the formation of pairs of electrochemical batteries, which in turn causes a direct reduction on the active site catalytic activity by accepting electrons [37,38].

**Table 3.** Activation energy of CIP removal under Fe-Cu and Fe-C treatment.

| Material | Ea (kJ/mol) |
|---|---|
| Fe-Cu | 14.93 |
| Fe-C | 16.87 |

Xu et al. (2008) [39] used Fe-Cu electrolytic materials—nitrobenzene (100 mg/L)—to treat an aqueous solution at the optimum pH range of 3–4, the weight of the internal electrolyte material of 50 g/L, treatment time of 90 min, achieving a nitrobenzene removal efficiency of 90%. The authors have fabricated internal electrolytic materials by two methods. In the first method, iron scrap (CT3 scrap) with a surface area of 0.3–0.4 $m^2$/g, washed with soap to remove oil, was used. The catalyst was pure copper foil with a thickness of 0.12 mm and was cut into $3 \times 1$ cm thin pieces before the experiments, mixed in the ratio Fe:Cu = 10:1 (mass). In the second method, scrap iron (CT3 scrap) with a surface area of 0.3–0.4 $m^2$/g, washed with soap to remove oil, was plated in a $CuSO_4$ solution with a concentration of 0.1 mol/L. For the Fe-Cu electrolytic material by the chemical plating method, after 60 min of treatment, the nitrobenzene removal efficiency reached 95%. This shows that the Fe-Cu electrolytic material fabricated by the chemical plating method are able to catalyze and create better reactions than internal electrolytic materials made by the mechanical mixing method of a mixture of scrap Fe and copper foil.

*3.8. Thermodynamics of Remove CIP*

The thermodynamic results of CIP removal are shown in Figure 9b and Table 4.

**Table 4.** Thermodynamic parameters for the removal of CIP by materials Fe-Cu and Fe-C.

| T (K) | $\Delta H^0$ (KJ/mol) | | $\Delta S^0$ (KJ/mol) | | $\Delta G^0$ (KJ/mol) | |
|---|---|---|---|---|---|---|
| | **Fe-Cu** | **Fe-C** | **Fe-Cu** | **Fe-C** | **Fe-Cu** | **Fe-C** |
| 298 | 2.493 | 2.494 | | | 73.639 | 72.538 |
| 308 | 2.576 | 2.578 | −0.239 | −0.235 | 76.110 | 74.972 |
| 323 | 2.700 | 2.702 | | | 79.816 | 78.622 |

The positive ΔH proves that CIP removal is endothermic. A negative entropy of −0.239 and −0.235 KJ/mol (which is near zero and is also relatively positive) indicates the rapid removal of the CIP molecules into the removal products [38].

**4. Conclusions**

The ciprofloxacin removal ability of Fe-Cu electrolytic materials and Fe-C materials was explored with respect to a number of experimental parameters. Fe-Cu was fabricated by the chemical plating method, and Fe-C materials were fabricated mechanically from iron powder and graphite. The considered conditions were pH, time, the mass of material, shaking speed, initial concentration of CIP and temperature. The results show that at a pH value of 3, shaking time of 120 min, shaking speed of 250 rpm, the mass of Fe-Cu and Fe-C material of 2 g/L, initial CIP concentration of 203.79 mg/L, resulted in a CIP removal efficiency of the Fe-Cu material of 90.25% and that of the Fe-C material of 85.12%. The removal of CIP of Fe-Cu and Fe-C materials follows pseudo-first-order kinetics. The activation energy of CIP removal of Fe-Cu material is 14.93 KJ/mol, that of Fe-C material is 16.87 KJ/mol. The positive ΔH proves that CIP removal is endothermic. A negative entropy of 0.239 and 0.235 kJ/mol (which is near zero and is also relatively positive) shows the rapid removal of the CIP molecules into the removal products.

Under the same reaction conditions, the CIP removal efficiency of Fe-Cu materials is higher than that of Fe-C materials. The results of using Fe-Cu and Fe-C internal electrolytic materials in CIP removal show that the material is capable of decomposing CIP at large concentrations with high efficiency.

From the above results, it is possible to combine Fe-Cu and Fe-C materials with biological methods to treat antibiotics in water environments.

**Author Contributions:** Writing—original draft preparation, T.H.D. and T.K.N.T.; data curation, X.L.H. and P.C.N.; conceptualization, P.C.N. and N.B.H.; methodology, N.B.H. and X.L.H.; formal analysis, T.T.A.D. and T.H.D.; writing—review and editing T.H.D., T.T.A.D. and T.K.N.T. All authors have read and agreed to the published version of the manuscript.

**Funding:** This research received no external funding.

**Institutional Review Board Statement:** Not applicable.

**Informed Consent Statement:** Not applicable.

**Data Availability Statement:** All the data is available within the manuscript.

**Conflicts of Interest:** The authors declare no conflict of interest.

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
