# Peer review of "Optimization, Kinetics, Thermodynamic and Arrhenius Model of the Removal of Ciprofloxacin by Internal Electrolysis with Fe-Cu and Fe-C Materials"

_processes, doi:10.3390/pr9122110_

Round 1
Reviewer 1 Report
Comments for Processes-1437798
Manuscript Number: Processes-1437798
Title: Optimization, Kinetics, Thermodynamic And Arrhenius Model Of The Decomposition Of Ciprofloxacin By Internal Electrolysis With Fe-Cu And Fe-C Materials
Article Type: Original Research Paper
In this paper Ciprofloxacin (CIP) removal ability of Fe-Cu and Fe-C electrolytic materials were examined with respect to pH, time, shaking speed, material mass, temperature and initial CIP concentration. The authors also studied kinetic, thermodynamic, and arrhenius model in the decomposition of Ciprofloxacin degradation with Fe-C and Fe-Cu electrolytic materials in aqueous medium.
This research has little innovation, in this regard, the paper is not acceptable for publication in its present form.
In the revision process, the following revisions should be pay attention.
- The innovative aspects of the paper are weak. There are many researchers studying Fe-Cu and Fe-C electrolytic materials.
- The authors studied CIP removal ability of Fe-Cu and Fe-C electrolytic materials separately when the conditions of pH, time, shaking speed, material mass, temperature and initial CIP concentration were changed. While the authors did not contract the differences of Fe-Cu and Fe-C materials, and did not tell us why they were different.
- The authors mentioned CIP degradation or decomposition many times, but according to the experiments, the CIP may be adsorbed or flocculated or degraded, just part of CIP was degraded. They could not say those effects as “CIP degradation or decomposition” but as “CIP removal”.
- The authors did not propose mechanism with enough tests and experiments. And what are the intermediate products and final products of CIP degradation?
- Numbers of grammatical and typographical mistakes are observed in the manuscripts. Therefore, the paper should be read throughout again and correct carefully. The examples are as follows:
Line 14 “…with respect to pH, time, shaking speed, material mass, temperature… ”, the ime means what time? Temperature means what time? It didn’t interpret that clearly.
Line 23 there was no unit behind the number 0.235.
Line 58 and 59, what is “Acit”?
Line 69 to 72, “If in the solution the organic substances RX (organochlorine compound), RNO2 (aromatic nitrocyclic compound) are components capable of accepting electrons on the anode surface (Fe) to transfer to the cathode, they are reduced according to the chlorination and aminoation reaction.” the organic substances RX and RNO2 are deduced according to de-chlorination and de-aminoation reactions, but not chlorination and aminoation reaction.
Line 69 and 77, “OH* radicals ” or “radicals OH* ” should be “・OH” or “hydroxyl radicals”.
Line 362, what is “vao”?
Too much to list them all.
Author Response
Dear Reviewers and Editors,
The answers to the questions you raised are detailed in the attached file below.

Reviewer 2 Report
- Please , improve the introduction.
- Add the following bibliographies between lines 31 to 38, to enrich the introduction. 10.1016 / j.eti.2021.101589, and 10.4067 / S0717-97072020000404943, 10.1016/j.envres.2019.109014, 10.1016/j.chemosphere.2020.127416
- Please reorder the information presented on lines 115 to 118 in table form and add more information on the antibiotic's portfolios, add references based on the information.
- Please modify the structure of the antibiotic (Figure 1), it is pixelated
- Please describe the antibiotic measurement conditions in figure 2, at what concentration the spectrum was taken, pH.
- Because, the authors did not show the characterization of the materials, although they cite the characterization of the materials in two manuscripts that the authors published. For this they must restrict it to a maximum of 8 to 9 figures in the whole document, or address it in materials supplementary
- On line 124-125, please change M for mol / L
- On line 144 please put the super index plus sign
- Please correct the chemical formulas present from line 98 to 102, check the entire document
- Line 97 check the degrees centigrade
- Please refer to the materials and methods section, section 2.3. What is the detection and quantification limit of the UV-Vis used?
- Please check the English of the legend of figure 13.
- Please, reduce the number of figures to 8 and tables to 5. You can group figures to avoid the document showing many figures
- please improve the presentation of the results in the tables presented
- please check the spacing from lines 31 to 55
- Because the reuse cycles of the materials present were not studied.
Author Response

(The authors gave the same response as above.)

Reviewer 3 Report
Please insert thermodynamic modelling section prior Materials and methods. This will expound on section 3.8. The explanation on effect of temperature on decomposition can be improved.... can authors insert schematic diagram for the decomposition.
Author Response

(The authors gave the same response as above.)

Round 2
Reviewer 2 Report
The authors satisfactorily addressed all the comments and suggestions mentioned, so I suggest publishing this manuscript.